# Race-Related Stress as a Driver of Postpartum Depression Among a Sample of Black Mothers

**DOI:** 10.3390/bs15111533

**Published:** 2025-11-11

**Authors:** December Maxwell, Ric Munoz, Sarah Leat, Corrina Jackson

**Affiliations:** 1Anne and Henry Zarrow School of Social Work, The University of Oklahoma, Norman, OK 73019, USA; rmunoz@ou.edu; 2School of Social Work, University of Memphis, Memphis, TN 38152, USA; srrbnsn2@memphis.edu; 3Supporting Healthy Initiatives for Tulsa, Tulsa, OK 74119, USA; cjackson@shift-tulsa.org

**Keywords:** race, perinatal mental health, stress, postpartum depression

## Abstract

In the US, research suggests that racial disparities exist in the prevalence of postpartum depression (PPD) and postnatal anxiety (PNA), with Black mothers experiencing PPD and PNA at a higher rate than their white counterparts. As a result, research that attempts to understand the antecedents of PPD and PNA in Black mothers may have value to the development of better interventions to reduce both in this subpopulation. Theory suggests that race-related stress (RRS) may be a contributing factor to PPD and PNA symptoms among Black mothers. RRS is defined as the stress associated with racism and discrimination encountered by Black women in their daily lives. In the current study, to test the relationship of RRS to PPD and PNA, we surveyed (*N* = 79) Black mothers who recently gave birth. The survey consisted of the Index of Race-Related Stress (IRRS), the Edinburgh Postnatal Depression Scale (EPDS), and the Postpartum Specific Anxiety Scale (PSAS-RSF), along with items capturing income, education, mental health status, and the number of children per mother. Income and mental health status, education, and the number of children per mother were used as covariates in a multivariate regression model with IRRS scores as the independent variable and EPDS and PSAS-RSF scores as twin dependent variables. These covariates were selected because of their established relationship with PPD and PNA. The data was analyzed using structural equation modeling. The results indicated that the model provided good fit to the data, (*X*^2^ = 6.32, *df* = 9; *p* = 0.707; RMSEA = 0.00 [90% CI: 0.000, 0.097]; CFI: 1.0). Moreover, IRRS scores were significantly correlated with both PPD symptoms (β = 0.45; *p* < 0.001) and PNA symptoms (β = 0.3837, *p* < 0.001), respectively. Such results suggest that future research into the role race-related stress plays in the development of PPD symptoms and PNA symptoms may have value in the reduction in both among Black mothers.

## 1. Introduction

PMADs (perinatal mood and anxiety disorders) have detrimental effects for the mother, the child, and the entire family system ([20]). PMADs include postpartum depression (PPD), generalized anxiety disorder (GAD), obsessive–compulsive disorder (OCD), and post-traumatic stress disorder (PTSD) ([14]). PPD is a serious mental health condition marked by ongoing sadness, anxiety, and fatigue after childbirth ([2]). PPD is a form of major depressive disorder, defined by the Diagnostic and Statistical Manual of Mental Disorders (DSM) as the onset of depressive symptoms following childbirth. These symptoms may include apathy, lack of interest in the baby, anxiety about bonding, feelings of inadequacy or guilt as a mother, persistent sadness, hopelessness, and fears of self-harm or harm to the baby ([2]). Postpartum anxiety (PPA) is a type of PMAD characterized by excessive worry, nervousness, or fear that occurs after childbirth. It often involves intrusive thoughts, physical symptoms (such as rapid heartbeat or dizziness), and a sense of dread, even when no immediate threat is present ([25]). Unlike the “baby blues,” postpartum anxiety is more intense and persistent, potentially interfering with a parent’s ability to care for themselves or their baby.

In the United States, PMADs are a significant contributor to maternal mortality, primarily through suicide, which is a leading cause of maternal death in the postpartum period ([16]). Annually, between 600,000 and 900,000 mothers experience PMADs in the US ([4]) and between 7 and 63% of mothers worldwide ([60]), likely an underestimate given the underdiagnosis of PMADs ([50]). Many deaths due to suicide occur within the first year postpartum, and a substantial portion are linked to untreated or undertreated PMADs like PPD and PPA ([16]).

In addition to the contribution to maternal deaths, PMADs have numerous other deleterious impacts on new mothers, infants, families, and communities. When left untreated, PPD can become a chronic condition, heightening a mother’s risk for future depressive episodes and negatively affecting her physical health ([68]). For infants, maternal PPD may disrupt early bonding and attachment, which can contribute to delays in emotional, cognitive, social, and physical development. Children of mothers with persistent PPD and PPA are at greater risk for behavioral problems, academic struggles, and long-term mental health issues. PMADs can also place significant strain on family dynamics—affecting the mother’s capacity to care for her child, contributing to early cessation of breastfeeding, and increasing the likelihood of depression in partners ([68]). The effects of PMADs often reach far beyond the postpartum period, shaping a family environment that may undermine both maternal health and healthy child development ([68]).

Black, Indigenous, and other women of color are disproportionately affected by both maternal mortality and inadequate mental healthcare, compounding the risk of fatal outcomes associated with PMADs ([15]). Furthermore, intersecting identities can compound the development of PMADs, such as disability status ([49]) and sexual orientation ([70]). For women of color, PMADs often go undiagnosed or underreported, due to stigma, lack of screening, and gaps in postpartum care ([73]). Additionally, larger systemic issues related to structural racism, cumulative stress, and medical gaslighting compound the effects of PMADs. Structural racism refers to the systemic, institutionalized policies, practices, and cultural representations that reinforce racial inequity ([11]). In the context of Black maternal health, it manifests in many ways, such as health care disparities. Black women in the U.S. are more likely to receive lower-quality prenatal and postpartum care, face dismissal of pain or symptoms by providers, and encounter implicit bias during medical interactions ([63]). Obstetric violence, a term increasingly used to describe the specific forms of racialized mistreatment Black women face during childbirth, including coerced medical decisions, lack of informed consent, and neglect, also contributes to structural racism ([30]). This mistreatment can be traumatic and increase the risk of developing postpartum mental health conditions. Additionally, structural racism is perpetuated through socioeconomic inequality, whereby generations of discriminatory practices (e.g., redlining, unequal education, employment discrimination) have led to disproportionate economic hardship and limited access to mental health resources ([31]; [24]).

Cumulative stress refers to the chronic, compounding exposure to stressors over time ([48]). For Black women, these include not only everyday life stressors (work, family, finances) but also race-related stressors such as microaggressions, vigilance and hyperawareness, and intergenerational trauma ([12]). Microaggressions are frequent, often subtle slights or insults that accumulate over time, eroding emotional well-being ([76]). Vigilance and hyperawareness refer to the need to constantly self-monitor and anticipate discrimination or bias in everyday settings, which places a constant psychological burden ([62]; [64]). The transmission of historical and lived racial trauma through family and community, or intergenerational trauma, can shape emotional resilience and stress responses ([62]; [64]). The weathering hypothesis, coined by Dr. Arline Geronimus, proposes that the chronic exposure to social, economic, and political exclusion causes early health deterioration in Black women ([32]). This “weathering” effect increases vulnerability to adverse outcomes like PPD, independent of socioeconomic status ([29]).

Finally, medical gaslighting, also often referred to as symptom dismissal, refers to situations where healthcare providers minimize, downplay, or dismiss a patient’s reported symptoms, often attributing them to psychological causes or suggesting they are exaggerating or misinformed ([22]). This experience can be deeply invalidating and lead to delayed diagnoses, inadequate care, or complete disengagement from healthcare systems. For Black patients and other people of color, medical gaslighting is not only a matter of individual provider behavior but is deeply intertwined with race-related stress and structural racism in healthcare. Medical racism has a long history, from unethical experimentation to widely held false beliefs (e.g., that Black people feel less pain) ([8]). These legacies persist in the form of implicit biases and under-treatment of pain or other complaints in Black patients today ([7]). This means symptom dismissal is often racialized, compounding the stress of both illness and discrimination.

These factors, structural racism, cumulative stress, and medical gaslighting, contribute to both underdiagnosis and undertreatment of PPD. Recent research indicates that the disparities in maternal mental health outcomes that disproportionately affect women of color are related to lifetime stress, which is compounded by stress brought on by racism ([21]). Given the profound effects of PPD on maternal and child health, it is critical to examine how sociocultural factors, such as race-related stress, may contribute to its onset and severity, especially among historically marginalized populations. In the US, research suggests that racial disparities exist in the prevalence of PMADs, with Black mothers experiencing PMADs at a higher rate than their white counterparts ([55]). As a result, research that attempts to understand the causes of PMADs in Black mothers may have value to the development of better interventions to reduce PMADs in this subpopulation.

### In the Literature

Race-related stress often goes unacknowledged in clinical settings, where symptoms may be misattributed or minimized. For instance, Black women reporting emotional distress postpartum may be dismissed or perceived as “resilient,” leading to underdiagnosis of PPD ([28]). There are studies surveying the relationship between living in disadvantaged neighborhoods (which can be a proxy for racialized stress but not a direct measurement) and PMAD for Black mothers ([55]), and linking experiences of emotional upset due to racism (EUR) to PPD using one variable to measure EUR ([10]), but, to date, no studies have evaluated Black women’s experiences with racial stress using the Index of Race-Related Stress (IRRS) as they relate to PMAD specifically.

## 2. Materials and Methods

### 2.1. Data Collection

The inclusion criteria involved women who identified as Black, could speak English, reported being at least 18 years old, and had given birth within the past 5 years. Although most studies of perinatal depression and anxiety follow mothers up to four years postpartum, we extended our inclusion window to five years. This decision was made to capture a broader sample and reflect the continued relevance of perinatal mental health concerns across the early parenting period, while still maintaining reasonable accuracy of recall. This longer window aligns with conventions in maternal and child health survey research, where five years is frequently used as a cutoff ([71]). We recognize that this may introduce some heterogeneity relative to studies with shorter follow-up periods, but believe it provides valuable insight into enduring associations between race-related stress and postpartum mental health. The larger sample came from a study investigating the relationship between various variables and maternal mental health outcomes and was *n* = 412 women who live in the United States and gave birth in the United States in the past five years. The subsample pulled for this study included only women who self-identify as Black, living in the United States, who met the inclusion criteria of having given birth within the past 5 years in the United States.

The study used Qualtrics recruitment and data collection services. Participants were recruited from various sources, including website intercept recruitment, member referrals, targeted email lists, gaming sites, customer loyalty web portals, permission-based networks, social media, etc. Participants were awarded compensation in the form of loyalty points (per their preference), and points varied based on their use of the survey software and completion of the survey. The survey included demographics, the Index of Race-Related Stress (IRRS; [75]), the Edinburgh Postnatal Depression Scale (EPDS; [19]) and the PSAS-B ([74]). IRB approval, including approval for the informed consent process of the study, was granted by the University of Oklahoma, IRB #17068.

### 2.2. Participants

All participants within the study identified as Black mothers. The average age of the sample was 32.27 years (SD = 8.76). The remaining sample demographics of the sample are reflected in Table 1.

### 2.3. Measures

#### 2.3.1. Index of Race-Related Stress

Individual differences in RRS were measured using the Index of Race-Related Stress Brief Form (IRRS-B; [74]). The IRRS-B is a validated self-report instrument that measures the frequency of respondents’ experiences of racism-related stress. The IRRS-B employs 22 items to capture experiences across three facets of racism: individual racism (e.g., interpersonal discrimination), institutional racism (e.g., systemic inequities in education, employment, or healthcare), and cultural racism (e.g., devaluation of cultural norms, heritage, or identity).

Higher scores of the IRRS indicate greater exposure to racial discrimination. The IRRS-B has demonstrated strong internal consistency and construct validity in Black populations ([74]). The IRRS-B is well established as a valid measure of RRS and continues to be regularly utilized in research ([17]; [53]). Total scores on the IRRS were used for analysis.

#### 2.3.2. Edinburgh Postnatal Depression Scale (EPDS)

To measure individual differences in symptoms of PPD, we employed the Edinburgh Postnatal Depression Scale (EPDS). The EPDS is the most widely used screening tool designed to identify symptoms of postpartum depression in new mothers. Developed in 1987, the EPDS consists of 10 self-reported questions that assess the emotional well-being of individuals during the postpartum period, focusing on mood, anxiety, and depressive symptoms ([19]). Each item is scored on a scale of 0 to 3, with higher scores indicating an endorsement of greater symptoms of depression. The tool is validated for use across diverse populations and has been translated into multiple languages, making it an essential resource for early detection and intervention in maternal mental healthcare. Total scores on the EPDS were used to capture variance in participants’ PPD symptoms.

#### 2.3.3. Postpartum Specific Anxiety Scale Research Short Form

To measure symptoms of PNA, we used the Postpartum Specific Anxiety Scale (PSAS-RSF) ([66]). This 12-item measurement evaluates anxiety symptomatology specific to the postpartum experience. The PSAS-RSF asks questions surrounding worry about the baby or parenting, as well as questions about parental efficacy and anxiety. Using a 0–4 Likert scale, the PSAS-RSF-C asks questions such as “I have felt unconfident or incapable of meeting my baby’s basic care needs.” Higher scores on the PSAS-RSF indicated a greater endorsement of symptoms of PNA. Scores on the PSAS-RF have shown good internal consistency and validity ([66]). Total scores on the PSAS-RSF were used for analysis.

#### 2.3.4. Covariates

In testing the model of IRRS-B scores as predictors of PPD symptoms and PNA symptoms, our analyses included four covariates control variables: (1) mental health diagnosis (coded as 0 if respondent reported no mental health diagnoses and coded as 1 if a participant reported a diagnosis); (2) income level (coded as 0 if respondent reported an income of <$45,000 and coded as 1 is the participant reported income > $45,000); (3) education level (coded as 0 if participant reported no college degree and coded as 1 if the participant reported a college degree; (4) the total number of children reported by each participant. All the covariate variables are self-reported by the participants.

Mental health diagnosis, income, education, and number of children were chosen as covariates because existing research suggests that mental health conditions ([41]; [61]), education ([77]), income level ([65]; [35]), and the number of children of a mother ([1]) are all associated with postnatal mental health variables such as PPD symptoms and PNA symptoms. Employing covariates in a regression model increases the power of the model ([59]) to detect significant correlations between selected independent variables and the dependent variables of a model.

#### 2.3.5. Data Analysis

Covariance-based structural equation modeling (CB-SEM) was used to test a model of RRS as a driver of the twin dependent variables of PPD symptoms and PNA symptoms among a sample of AA mothers (Figure 1). We selected CD-SEM as our data analysis tool because CB-SEM allows for the modeling of our twin dependent variables within the same statistical model, reducing the likelihood of type I error ([40]). All calculations utilized maximum likelihood (ML) estimations performed with the SPSS Amos add-on ([3]).

Per standing CB-SEM modeling practice ([40]), we employed multiple fit tests to evaluate the proposed theoretical model. First, employed a *X*^2^ test with a threshold of *p* > 0.05 indicating acceptable fit ([40]). We also utilized additional fit tests that included the Comparative Fit Index (CFI) with a threshold of ≥0.90 as an indication of acceptable fit ([6]; [38]). We also employed the Root Mean Square Error of Approximation (RMSEA), with a threshold of ≤0.10 indicating acceptable fit ([13]).

#### 2.3.6. Bootstrapping

To increase the strength of our conclusions regarding the generalizability of the variable relationships identified within the sample, we employed bootstrapping resampling. With bootstrapping, subsamples are randomly drawn, with replacement, from the original sample. Each subsample is then used to establish a point estimate of the variable relationships in the population. This process is repeated many times, with an *N* = 5000 often referenced as the minimum number of resamples to execute ([34]). When the CI interval of the point estimate does not contain the value of 0, the relationship is considered statistically significant ([34]).

#### 2.3.7. *f*^2^ Test

To measure the impact of RRS on the twin variables of PPD symptoms and PNA symptoms, we also utilized the *f*^2^ test. The *f*^2^ test is an effect size measure that operates by evaluating how much variance is accounted for in dependent variables, in this case, PPD and PNA, when a specific independent variable is included in a model. In this case, we used the *f*^2^ test to examine how much variance was accounted for in PPD symptoms and PNA symptoms when RRS scores were included in the model.

## 3. Results

To begin, the mean score of the EPDS was *µ* = 13.9 while the mean score of the PSAS-RF was *µ* = 14.6. We began hypothesis testing by examining the normality assumptions of ML analysis ([40]). The results of normality testing indicated that all items exhibited normality. We also evaluated the internal consistency of the IRRS (α = 0.946), the EPDS (α = 0.896), and the PSAS-RSF (α = 0.911) and found all to have scored above accepted thresholds. Additional descriptive statistics of the scale values are included in Table 2, with the zero-order correlations between variables captured by Table 3.

The initial testing of the proposed model indicated that the model produced a poor fit (X). An examination of the modification indices (MI) indicated that the good fit would result if the error terms of the income and education variables were allowed to correlate. Correlating error terms is supported when the indicators in question are theoretically linked ([58]). In this case, income and education are theoretically both indicators of the underlying latent variable of socio-economic status.

Upon adjusting the model to reflect the relationship of income and education as variables of socio-economic status, the results of testing the proposed model indicated that the model exhibited excellent fit (*X*^2^ = 6.32, *df* = 9; *p* = 0.707; RMSEA = 0.00 [90% CI: 0.000, 0.097]; CFI: 1.0). Moreover, an examination of the standardized beta values revealed that IRRS scores exhibited moderate ([18]) positive correlations with PPD symptoms (β = 0.45; *p* < 0.001) and PNA symptoms (β = 0.37, *p* < 0.001), respectively. The overall model, including the covariates, accounted for an *R*^2^ = 0.291 of variance in PPD symptoms and an *R*^2^ = 0.201 of variance in PNA symptoms.

To test the stability of the model in the population, we next performed a bootstrapping test. The results indicated that after *N* = 5000 resamples, the 95% CI of the point estimate of the relationship between IRRS scores and PPD symptoms did not contain the number 0 (β = 0.451; 95% CI: [0.297; 0.593]) nor did the point estimate of the relationship between IRRS scores and PNA symptoms (β = 0.375; 95% CI: [0.190; 0.535]). Such a result further indicates that the variable relationships are statistically significant and adds evidence of the stability of these relationships in the parent population of Black mothers.

Finally, to empirically partition the contributions of RRS to PPD symptoms of PPD and PNA, we conducted the *f*^2^ test. An examination of the *f*^2^ values of the model indicated that the presence of RRS accounted for an *f*^2^ = 0.242 in PPD symptoms and *f*^2^ = 0.164 in PNA symptoms. According to accepted heuristics, such *f*^2^ values are moderate in magnitude ([18]).

## 4. Discussion

Despite growing recognition of racial disparities in maternal mental health, the role of race-related stress remains underexamined in statistical models of postpartum depression (PPD) and postpartum anxiety (PPA). Our findings challenge this oversight. In a sample of Black postpartum women, we found that race-related stress, as measured by IRRS scores, was a predictor of both PPD symptoms and PNA symptoms, even after controlling for other known risk factors such as mental health diagnosis, income, and total number of children per participant. This suggests that the psychological toll of racism and discrimination is not only socially consequential but statistically robust in shaping postpartum mental health outcomes. Moreover, the mean scores on the EPDS (*µ* = 13.9) suggest that this sample of African American mothers is experiencing depressive symptoms at a relatively high rate according to established scoring norms. To wit, general cutoff scores for the EPDS suggest that scores of 13 or higher indicate likely postnatal depression, with scores of 10 or higher indicating the need for further clinical assessment ([43]). For the PSAS, the mean score (*µ* = 14.6) of the sample indicates the sample had a mean score lower than the threshold suggested for a clinical anxiety designation, e.g., a PSAS score of 26 ([47]).

Such a finding is significant because while race is often included as a demographic descriptor, it is rarely operationalized in ways that capture the lived impact of racial stress. Our study takes an important step toward quantifying the impact of racism, via the psychological state of racism related stress, and centering it within perinatal mental health research. Previous studies support these findings, which indicate that racism-related stress, including daily microaggressions and overt discrimination, is linked to increased rates of perinatal posttraumatic stress disorder (P-PTSD) among Black mothers ([39]). Experiences of disrespect and abuse by perinatal care providers can amplify PTSD symptoms in the postpartum period ([39]). Although P-PTSD and PPD and PPA differ, there is strong comorbidity among the three ([27]; [5]; [54]; [23]), and highlights the link between race-related stress and perinatal mental health outcomes for Black mothers.

The results of this study point to a critical and often overlooked driver of postpartum psychological distress: race-related stress. This form of chronic psychosocial stress, rooted in direct experiences of racism, structural inequities, and vicarious racial trauma, appears to exacerbate the risk for both anxiety and depression following childbirth. That these effects persisted after controlling for socioeconomic and interpersonal variables reinforces that the experience of racism is not reducible to poverty or lack of access but is an independent and compounding source of stress. Indeed, the previous literature indicates that chronic exposure to race-related stress and adversity alters maternal physiology ([9]; [36]), increasing inflammation and stress hormones ([33]), which are linked to adverse birth outcomes such as preterm birth and low birthweight ([45]). These biological effects may also be transmitted across generations, perpetuating health inequities.

Importantly, these findings align with emerging frameworks that conceptualize racism as a form of toxic stress with biological and psychological consequences. Prior research has established that chronic stress exposure during the perinatal period contributes to dysregulated cortisol levels, sleep disruption, and poor emotional regulation ([56]), all of which are associated with heightened risk for PPD and PPA. Our data extend this literature by showing that racism-related stress specifically may serve as a hidden but potent contributor to these outcomes in Black women. Moreover, these results call attention to the inadequacy of mental health models that treat all postpartum people as equally vulnerable, ignoring how sociopolitical context shapes risk. Standard screening tools for perinatal mental health rarely assess racial trauma, and clinical care pathways often fail to validate or address these experiences. By statistically confirming the contribution of race-related stress to postpartum mood and anxiety disorders, our findings make a compelling case for integrating assessments of racism into both clinical care and epidemiological research on maternal mental health.

### 4.1. Implications for Practice and Policy

The findings of this study highlight numerous implications for practice and policy. The link between race-related stress and PMAD indicates a need for culturally responsive screening tools. The findings also illuminate the need for the creation and use of by clinicians, of perinatal-specific mental health screeners that explicitly include items addressing racial stress and discrimination, as standard tools may overlook these key factors. Additionally, integrated care models could address some of the disparities arising from race-related stress in the perinatal care environment ([37]), and perinatal care should include embedded behavioral health services that are trained in racial trauma and culturally informed care for Black women ([57]). In addition to the relationship examined in this study (race-related stress and postpartum mental health), an increasingly robust body of literature highlights the role of community-based, culturally congruent support interventions, particularly the use of doulas, as a promising strategy for mitigating the effects of racism, discrimination, and inequities in the perinatal period for Black women. For example, doula care has been associated with lower odds of preterm birth and caesarean delivery, improved breastfeeding initiation and continuation, and greater reports of respectful care, effects that are especially pronounced among non-Hispanic Black birthing individuals ([69]; [52]; [46]; [26]). Qualitative work suggests that doulas may buffer the impact of institutional racism by serving as advocates, aligning care with cultural values, and providing emotional and informational support during pregnancy, birth, and the postpartum period ([72]). Although our current analysis did not directly test a doula-intervention effect on postpartum depression, acknowledging this literature helps place our findings in a broader context of culturally relevant strategies for improving maternal mental health and reducing disparities. Future research could integrate doula support as a moderator or mediator of the connection between race-related stress and PPD, explore mechanisms (e.g., increased respectful care, reduced perceived racial discrimination, enhanced social support), and test scalable models targeted to Black women experiencing high levels of race-related stress. Given the understandable distrust of medical providers by Black women, future research should explore whether community-based supports, such as culturally grounded support groups and doulas rooted in Black communities, may help buffer the effects of racial stress and provide contextually appropriate care.

These findings also highlight the need to protect race-based data collection. The link between race-related stress and PMAD affirms the need for continued race and ethnicity data in health research and surveillance systems ([42]; [37]). Policies that limit or erase these variables undercut evidence-based care. In addition, targeted funding is needed to address such disparities. Continued and expanded funding for maternal mental health initiatives must explicitly prioritize Black women and address systemic racism as a social determinant of health. Given that disparities vary by state, states and federal agencies should require racial equity impact assessments in perinatal mental health initiatives, similar to environmental impact assessments.

### 4.2. Implications for Research

The need for longitudinal studies assessing how race-related stress contributes to chronicity or recurrence of PPD/PPA across the perinatal timeline and early parenting years is perhaps the most notable implication for research from these findings. Furthermore, future research should encourage research designs that include the intersections of race, gender, class, and historical trauma to capture the complexity of Black maternal mental health. Additionally, studies evaluating the impact of community-based perinatal care models (such as support groups, peer support, and doulas) can evaluate how much, if any, these models reduce the effects of race-related stress on PMAD.

In addition, our study relied on the Index of Race-Related Stress (IRRS), which captures multiple levels of racism, including individual, cultural, and institutional forms. Each construct has potential relevance for perinatal mental health. Individual racism (e.g., interpersonal discrimination or microaggressions) may exacerbate stress during pregnancy by heightening vigilance and eroding social support ([67]). Cultural racism (e.g., negative stereotypes, invisibility of Black women’s perinatal experiences) ([51]) can undermine women’s sense of belonging in clinical encounters, contributing to mistrust of providers. Institutional racism (e.g., differential treatment in healthcare settings, barriers to equitable access) has structural consequences that can limit timely and appropriate care ([33]). Considering these domains together provides a more comprehensive understanding of how racism operates across levels to influence postpartum depression risk.

The IRRS offers important strengths as a validated, multidimensional measure of race-related stress with strong psychometric properties. However, it also has limitations. As a self-report instrument, responses may be influenced by recall or social desirability bias, and the measure was not designed specifically for the perinatal context. Moreover, while it captures broad experiences of racism, it may not fully reflect intersectional stressors or unique perinatal encounters, such as provider communication around childbirth or postpartum care. Future work could adapt or supplement these measures to better capture perinatal-specific mechanisms of racial stress.

To expand on specific constructs of racial stress, we also suggest that future research consider the role of allostatic load and toxic stress in the relationship between race-related stress and perinatal mental health. Allostatic load, or the cumulative wear and tear on physiological systems resulting from chronic stress exposure ([32]), may provide a biological mechanism linking experiences of racism to heightened vulnerability for PMADs. Toxic stress, particularly when stressors are severe, prolonged, and occur without adequate buffering supports, can disrupt neuroendocrine and immune functioning in ways that are especially consequential during the perinatal period. By integrating these frameworks, future studies can examine how repeated exposures to race-related stress contribute to physiological dysregulation that, in turn, increases risk for postpartum depression and anxiety. Such research would strengthen understanding of the biopsychosocial pathways involved and inform interventions that address both social determinants and physiological sequelae of racism in the perinatal context.

### 4.3. Limitations

While this study offers insight into a potential contributor to PPD symptoms and PNA symptoms among Black women, the study is not without possible limitations. First, the data collection method was cross-sectional in nature. While we based our statistical model on a priori theory, considered the best practice when using cross-sectional data to test causal theories ([40]), it is possible that psychological states such as PPD symptoms and PNA symptoms drive increases in RRS among Black mothers rather than vice versa. While theory does not support this direction of causality, such a conclusion cannot be ruled out from cross-sectional data. Future research is needed to strengthen conclusions about the directionality of the variable relationships found in the current sample. Second, while bootstrapping was used to validate the stability of the variable relationships in the population, additional sampling would have value in furthering our conclusions about the relationship of RRS to postpartum mental health. Third, since all data was collected via a survey of self-report data, the presence of a common biased method cannot be ruled out. Nevertheless, despite potential limitations, the current study offers a new way forward to better understanding how RRS may contribute to PPD and PNA among Black mothers. Finally, although there is some evidence from maternal health literature that maternal recollection of childbirth experiences and feeding behaviors retains moderate consistency over multiyear intervals (e.g., [71]; [44]), recall of psychological symptoms or stressors may degrade more rapidly. Therefore, our 5-year inclusion window may introduce additional measurement error or attenuation of observed associations.

## 5. Conclusions

Research supports that variables such as PPD symptoms and PNA symptoms are higher in Black mothers relative to the general population. The current study provides a possible explanation for this, that Black mothers experience RRS in a manner that contributes to greater PPD symptoms and PNA symptoms. Should future research align with the current results, developing clinical interventions that reduce RRS may have the added value of reducing PPD symptoms and PNA symptoms among Black mothers.

## Figures and Tables

**Figure 1 behavsci-15-01533-f001:**
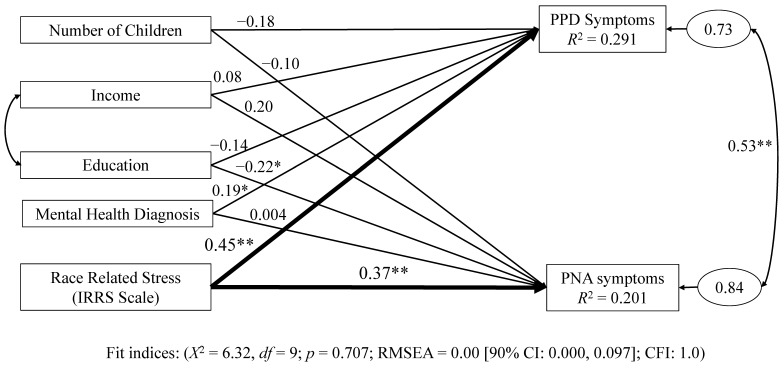
CB-SEM Model of IRRS Scores as a Predictor of PPD Symptoms and PNA Symptoms Among African American Mothers using Standardized Values (*N* = 79). Notes—* *p* < 0.05; ** *p* < 0.001.

**Table 1 behavsci-15-01533-t001:** Sample Demographics (*N* = 79).

	Raw Total	% of Total (Rounded)
Income (USD)		
0–19,000	13	17
20,000–45,000	27	34
46,000–75,000	18	23
76,000–100,000	15	19
101,000–150,000	4	5
Above 150,000	2	3
Mental Health Diagnosis		
Yes	28	45
No	51	55
Number of Children		
1	32	1
2	22	41
3	18	28
4	5	6
5	1	1
Education		
Some high school	22	28
Completed high school	2	3
GED	16	20
Some college	16	20
Associates degree	19	24
Bachelors degree	2	3
Master’s Degree	1	1
Doctoral Degree	1	1

**Table 2 behavsci-15-01533-t002:** Selected Descriptive Statistics for IRRS, EPDS, and the PSAS-RSF (*N* = 79).

Scale	Minimum Score	Maximum Score	Mean	SD	Skewness	Kurtosis
IRRS Scores	0	81	32.4	21.6	−0.090	−0.944
EPDS Scores	0	27	13.9	6.6	−0.131	−0.552
PSAS-RSF	0	34	14.6	9.0	−0.110	−0.927

**Table 3 behavsci-15-01533-t003:** Zero Order Correlations (*N* = 79).

Variables	1	2	3	4	5	6
1. Number of Children	1.96 (1.03)					
2. Income	−0.049	2.70 (1.24)				
3. Mental Health Diagnosis	−0.128	−0.011	0.35 (0.48)			
4. IRRS Scale (Race-Related Stress)	−0.106	0.006	0.093	32.4 (21.6)		
5. PPD Scale (Postpartum depression)	−0.233 *	−0.018	0.244 *	0.479 **	13.9 (6.63)	
6. PSAS Scale (Postnatal anxiety)	−0.112	0.113	0.029	0.385 **	0.596 **	14.6 (9.0)

Notes—* *p* < 0.05; ** *p* < 0.001. Means and standard deviations are posted along the diagonal.

## Data Availability

The datasets presented in this article are not readily available because the data are part of an ongoing study. Requests to access the datasets should be directed to December Maxwell.

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
