# Peer review of "Race-Related Stress as a Driver of Postpartum Depression Among a Sample of Black Mothers"

_behavsci, 2025, doi:10.3390/bs15111533_

Round 1
Reviewer 1 Report
Comments and Suggestions for Authors
Thank you for the opportunity to review this manuscript. This study provides valuable insight into race-related stress (RRS) as a significant risk factor for perinatal mood and anxiety disorder (PMAD) symptomology among Black women. The authors present a theory-driven analysis with significant results from a survey of 79 Black mothers. Overall, the manuscript is well-written and thoughtfully organized; however, there are areas within the introduction, methods, results, and discussion sections that would benefit from additional clarification and detail. Please find below specific feedback and recommendations:
Introduction
- The introduction could benefit from briefly acknowledging how other forms of discrimination, beyond racism, have been explored in the literature in relation to PMADs. Incorporating an intersectional perspective such as the impact of gender, disability status, weight, and sexual orientation. While this is briefly mentioned in the discussion regarding implications for future research, it would be helpful to introduce and acknowledge this relevant body of literature earlier in the manuscript.
Methods
- If the sample consists exclusively of indigenous birthing persons, provide a brief description of this population in the methods section.
- Were there any exclusion criteria for the sample?
- Report alpha values for the IRRS, EPDS, and PSAS-RSF scales to establish reliability within the present sample.
- Clarify if total scores were used for all measures (e.g., IRRS, EPDS, PSAS-RSF). Additionally if symptom scores were used this should be defined as such rather than a diagnosis throughout the manuscript.
- Specify how income was treated in the analysis (e.g., continuous or categorical variable).
- Consider providing a justification for why education was not included as a covariate if the data were collected, as it is an established risk factor for PMAD symptoms and could potentially influence the relationship between race-related stress and postpartum depression.
Results
- Include descriptive statistics for the independent and dependent variables (e.g., IRRS, EPDS, PSAS-RSF), which could be added to the demographic table (Table 1).
- Consider providing severity scores for the EPDS and PSAS-RSF to better contextualize symptom severity within the sample. While the authors report self-reported mental health diagnoses, incorporating severity information could enhance the discussion of generalizability by illustrating how the observed symptom levels compare to those in broader populations.
- Present a table or supplementary table with both univariate and multivariate statistical model results for clarity for readers.
Discussion
- If the sample is 100% indigenous women, discuss the generalizability of findings in the context of this population and how this may also be a strength/unique aspect of the present study.
- Provide citations for claims made in the implications section regarding policy and practice. For example, if discussing tools to assess RRS or measures to integrate these into provider workflows, cite supporting research.
- Include references for community-based strategies, such as the use of doulas, as there is literature supporting their role in improving maternal outcomes.
- Expand on implications for future research by further discussing concepts introduced in the introduction, such as allostatic load and toxic stress, and their relevance to RRS and PMADs.
- Remove subjective language (e.g., “we think”)
- Consider adding a limitation regarding potential recall bias, as the five-year postpartum timeframe may introduce variability in participants’ recollection of experiences in relation to their PMAD symptoms.
Reviewer 2 Report
Comments and Suggestions for Authors
Manuscript Summary: The investigators present findings and interpretations from a survey evaluation to assess the relationship between race related stress, symptoms of postpartum depression, and symptoms of postpartum anxiety. Black and African mothers were asked to respond to several validated measures, including the Index of Race Related Stress (IRRS), the Edinburgh Postnatal Depression Scale (EPDS), and the Postpartum Specific Anxiety Scale (PSAS). In the study, the authors accounted for income and mental health status, given their role as biasing variables in the evaluation. Findings demonstrate a positive correlation between IRRS scales and 1) postpartum depressive symptomology and 2) postpartum anxiety symptomology. The authors suggest that these results are a vital step to quantifying the crucial role of chronic stress on mental health following childbirth. Implications point to the need for implementation of community-based supports, changes to the perinatal care environment to be more responsive to racial trauma and culture, along with a need for continued race-based data collection. I commend the authors for a very comprehensive overview and context for the evaluation of this important public health issue. This work has the potential to catalyze ongoing research opportunities to support equitable perinatal mental health care.
Overall, I’m struggling to understand the extent to which the race related stress measured in the study comes from interactions across the perinatal period versus those that happen across the life course. This has important implications for intervention development to support equitable mental health in the perinatal period, specifically. The submitted manuscript also requires substantial clarification and organization of ideas in several areas, leading to major revisions at this stage. Detailed comments are provided below, with a particular focus on the methods and interpretations, which are underdeveloped and, in some cases, overstated. Thank you for this important work.
- INTRODUCTION
- Overall: The introduction has a lot of great and relevant information. This information can also be streamlined to enhance readability and reduce redundancy. For example, Lines 34 and 38 both state the same prevalence of PMADs in the US.
- Overall: You do not need to define the abbreviations throughout, just the first occurrence (ex: PMADs, PPA, PPD).
- Overall: Please include the s when using “PMADs.” It is often missing and should be included to describe the larger group of conditions, as insinuated here.
- Overall: If the goal of this paper is to discuss how racism, operationalized through race and chronic stress, impacts the perinatal mental health of Black and other racially minoritized groups, there needs to be much more engagement with current literature (or lack thereof) specific to PMADs and racial inequities. Many of the statements describing the adverse impact of PMADs or care in the perinatal period seem generalizable to all groups.
- Lines: 50-51: Please revise this statement. PMADs and SUD can co-occur, but they are not the same. PMADs are indeed complex and often debilitating conditions. However, they do not typically result in overdose as this sentence suggests. I think the language around substance use disorders should be removed from the manuscript altogether, unless deemed necessary. Currently, it may be more of distraction than adding substantive content to the goal of the paper. I encourage the authors to rethink the utility of including it.
- Overall: The introduction may benefit from restructuring to guide the reader along. Here is one suggestion:
- Paragraphs 1 and 2: PMADs are defined as _____, and inclusive of X conditions, 2) PMADs reflect a growing complex issue, and 3) PMADs contribute to poor maternal and infant outcomes including _____+__.
- Paragraph 3: Black and African American, along with other women of color, are disproportionately impacted by the adverse effects of untreated PMADS. Include relevant and recent epidemiologic data to support this.
- Paragraph 4: Women of color are also disproportionally impacted along the perinatal mental health care pathway, including inequities in screening and access to treatment due to XYZ (can include some of the existing content in section 1.1.3 here)
- Paragraph 5: Language around the role of structural racism, cumulative stress, and race related stress (essentially a streamlined discussion of sections 1.1, 1.1.1, and 1.1.2). In your discussion of structural racism and the cumulative stress, it would also be helpful to highlight how these constructs have been measured traditionally and any limitations of those approaches.
- Paragraph 6: Gaps in the literature and objectives of the evaluation (section 1.1.4)
- MATERIALS AND METHODS
- Overall: This section would benefit from additional details about the study population and design to support the reproducibility of study activities.
2.1 Data Collection
- Was this a U.S.-based sample? I suspect yes, but currently this is unclear. Which geographic region(s) did participants derive from?
- Were all participants born in the U.S? This has implications for how individuals understand race and experiences of racism.
- Did the study population self-identify as women, or was this assumed?
- Did all individuals identify as Black? Were African American-identifying individuals included? Typically, the literature is more inclusive and suggests “Black and African American” as there are people who may identify with one and not the other.
- Why did you choose individuals who had given birth in the past 5 years? Most of the literature thus far on perinatal depression and anxiety does not go beyond 4 years. Additional clarification would be helpful to contextualize.
- Did IRB approval include informed consent?
- Did participants need to speak English?
- Did participants themselves complete the survey, or did researchers administer the questions to the participants? What potential biases could arise from either of these approaches?
- On average, how long did it take for individuals to complete the surveys/measures?
- Were participants paid to participate?
- Were there any exclusion criteria?
- Please describe any screening procedures utilized to arrive at the 79 respondents in your sample.
2.2 Participants
- What group(s) of people does 'Indigenous’ refer to, and why is this important in the context of this study?
- Table 1:
- I’m assuming income is measured in USD. Please clarify.
- What types of mental health diagnoses were included in this binary
measure? Was this captured? Would it matter whether this diagnosis preceded the perinatal period?
- A footnote should be included if you do not plan to include it in the methods.
- Measures
2.3.1:Index of Race Related Stress
- The citation for the use of the IRRS is from 1996. Is there anything more recent about its use? This is important considering how we understand, and measure racism and chronic stress has changed dynamically over the last three decades.
- How many items are in the IRRS?
- You mention that higher scores indicate greater exposure to racial discrimination. What is the range of scores?
2.3.2 and 2.3.3: Edinburgh Postnatal Depression Scale and Postpartum Specific Anxiety Scale Research Short Form
- Sections regarding the EPDS and PSAS-RSF measures would benefit from additional description of the cutoffs associated with these tools. For example, what is considered mild, moderate, or severe symptomology?
- When is a clinical follow-up assessment typically indicated based on the literature?
2.3.4. Covariates
- Is the variable for any mental health diagnosis captured throughout the life course or at a particular period?
- There is recent literature that looks at income and the development of mental health conditions, some of which have contradictory results from those stated here.
- RESULTS
- Overall: Please revise throughout to clarify that it is symptoms of PPD and PPA that are being measured, not diagnoses of either of these conditions.
- What does ML stand for in Line 265?
- Line 282-283 could be clearer. Do you mean RRS accounted for a .242 in the variance of PPD symptomology?
- DISCUSSION
- Overall: The discussion would benefit from additional engagement with literature. I was surprised to see a substantial lack of citations throughout. For example, statements like those in 307-309 need to be cited.
- Overall: There appear to lot of bold claims about what the results of the study indicate, particularly given these correlations were on the cusp of moderate. I worry that several points in the discussion here overstate the findings. I won’t highlight each instance, but I would recommend tempering the language throughout, given the considerations associated with study design and sample size. One example of this includes statements such as those in lines 323-324.
- While the study does demonstrate a moderate correlation between the variables, please note that this correlation still does not tell us much about functionality and intensity. The IRRS is also not unique to the perinatal period. therefore, statements such as those on 336-337 should be revised.
- I would suggest revising this discussion to focus on the implications for the evaluation at hand. I was surprised to see mention for of community-based perinatal care models included in the author’s implications for research. This was not a focus of the paper and was not indicated based on these findings alone.
- The discussion needs more engagement with the components of the tools and measures themselves. For example, if the IRRS measures several components of racism. There remains an unharnessed opportunity to delineate the role of each construct (individual, cultural, and institutional racism) and the mechanisms by which they impact overall perinatal mental health symptom development and care. What are the strengths and limitations of these tools as measures?
- CONCLUSION
- The following statement, “….that Black mothers experience RRS in a manner that contributes to greater PPD and PNA” in Lines 382-384, may be overstated. This study demonstrates that there is a moderate correlation between these variables. I would reframe, emphasizing the value of further investigations to examine these associations, accounting for other mediating variables.
ABSTRACT:
- The E in EPDS is spelled incorrectly here. Please replace with “Edinburgh.”
Round 2
Reviewer 1 Report
Comments and Suggestions for Authors
Thank you for the opportunity to review this revised manuscript. The authors have made significant efforts to address the comments raised during the initial review, and these revisions have strengthened the paper considerably. The paper is well written and organized and contributes to the literature. The current methods and analyses are sufficient, and additional tables are not necessary given the updates provided. In future submissions, including a table summarizing the revisions with corresponding line numbers would help facilitate understanding how specific comments were addressed. While the manuscript has been improved overall, there are still a few minor revisions that require attention. Thank you for your efforts on this revision and your valuable contribution to the field of postpartum mental health equity.
- Abstract: Please correct the spelling of “Edinburgh” in reference to the EPDS.
- Methods: fix Takehara citation was there meant to be another citation here?
- Comment 10 (Symptom severity scores for outcome variables PPD and PNA): Including a discussion of the EPDS or PSAS scores would provide valuable context regarding the distribution of psychological distress symptoms in the sample, improving the interpretability and generalizability of the findings. The results skip to regression results instead of describing the outcome variables descriptives. For instance, noting that an average EPDS score of 13.9 reflects moderate depressive symptoms would highlight the severity, which is notably high for a community-based sample and merits acknowledgment.
- Comment 14 (Doula literature): There is substantial evidence demonstrating the positive impact of doulas in tailored interventions, particularly for Black women, in addressing race-related stressors and improving maternal outcomes during the perinatal period. Framing this as a future area of research overlooks the robust existing literature supporting this community-based strategy. Although outside the scope of this current analysis, acknowledging or incorporating this body of work into the discussion would strengthen the discussion and provide a more comprehensive perspective on culturally relevant interventions.
Author Response
Abstract: Please correct the spelling of “Edinburgh” in reference to the EPDS. This has been fixed.- Methods: fix Takehara citation was there meant to be another citation here? We have corrected this, we just had copied the citation from another part of the manuscript and forgot to put in the parentheses.
- Comment 10 (Symptom severity scores for outcome variables PPD and PNA): Including a discussion of the EPDS or PSAS scores would provide valuable context regarding the distribution of psychological distress symptoms in the sample, improving the interpretability and generalizability of the findings. The results skip to regression results instead of describing the outcome variables descriptives. For instance, noting that an average EPDS score of 13.9 reflects moderate depressive symptoms would highlight the severity, which is notably high for a community-based sample and merits acknowledgment.
Thank you for this. We have added language about cutoffs scores for both EPDS and PSAS-SF-C.
- Comment 14 (Doula literature): There is substantial evidence demonstrating the positive impact of doulas in tailored interventions, particularly for Black women, in addressing race-related stressors and improving maternal outcomes during the perinatal period. Framing this as a future area of research overlooks the robust existing literature supporting this community-based strategy. Although outside the scope of this current analysis, acknowledging or incorporating this body of work into the discussion would strengthen the discussion and provide a more comprehensive perspective on culturally relevant interventions.
Yes we agree. It was not our intention to frame it as such, more so we meant specifically as a path analysis/buffer. We have updated it to be more clear and more supportive of the existing literature.